# Image-Based High-Throughput Phenotyping of Cereals Early Vigor and Weed-Competitiveness Traits

**Shlomi Aharon [1,2], Zvi Peleg [2], Eli Argaman [3], Roi Ben-David [4] and Ran N. Lati [1,*]**

1   Department of Plant Pathology and Weed Research, Newe Ya'ar Research Center,
    Agricultural Research Organization (ARO)-Volcani Center, Ramat Yishay 30095, Israel;
    Shlomia@volcani.agri.gov.il
2   The Robert H. Smith Institute of Plant Sciences and Genetics in Agriculture,
    The Hebrew University of Jerusalem, Rehovot 7610001, Israel; Zvi.Peleg@mail.huji.ac.il
3   Soil Erosion Research Station, Ministry of Agriculture and Rural Development,
    Rishon LeZion 7528809, Israel; eliar@volcani.agri.gov.il
4   Institute of Plant Sciences, Agriculture Research Organization (ARO)-Volcani Center,
    Rishon LeZion 7528809, Israel; roib@volcani.agri.gov.il
*   Correspondence: ranl@volcani.agri.gov.il; Tel.: +972-50-220403

**Abstract:** Cereals grains are the prime component of the human diet worldwide. To promote food security and sustainability, new approaches to non-chemical weed control are needed. Early vigor cultivars with enhanced weed-competitiveness ability are a potential tool, nonetheless, the introduction of such trait in breeding may be a long and labor-intensive process. Here, two image-driven plant phenotyping methods were evaluated to facilitate effective and accurate selection for early vigor in cereals. For that purpose, two triticale genotypes differentiating in vigor and growth rate early in the season were selected as model plants: X-1010 (high) and Triticale1 (low). Two modeling approaches, 2-D and 3-D, were applied on the plants offering an evaluation of various morphological growth parameters for the triticale canopy development, under controlled and field conditions. The morphological advantage of X-1010 was observed only at the initial growth stages, which was reflected by significantly higher growth parameter values compared to the Triticale1 genotype. Both modeling approaches were sensitive enough to detect phenotypic differences in growth as early as 21 days after sowing. All growth parameters indicated a faster early growth of X-1010. However, the 2-D related parameter [projected shoot area (PSA)] is the most available one that can be extracted via end user-friendly imaging equipment. PSA provided adequate indication for the triticale early growth under weed-competition conditions and for the improved weed-competition ability. The adequate phenotyping ability for early growth and competition was robust under controlled and field conditions. PSA can be extracted from close and remote sensing platforms, thus, facilitate high throughput screening. Overall, the results of this study may improve cereal breeding for early vigor and weed-competitiveness.

**Keywords:** crop height; crop volume; drone; food security; three-dimensional model; MVS; projected shoot area; SFM

## 1. Introduction

Weeds are the major biotic stress constraining agricultural cereal crop production [1], directly (i.e., chemicals and other management inputs and labor cost) and indirectly (competition on resources, host to pests, reducing the yield quality, and increasing the cost of processing). Integrating cereal cultivars with ameliorate weed-competitiveness is a promising approach for sustainable weed

control by suppressing weeds establishment and reduce dependence on herbicides, hence decreasing selection pressure for weed resistance [2,3]. Wheat cultivars with enhanced weed-competitiveness traits have been demonstrated to effectively reduce the number of weeds and chemical applications (e.g., [4,5]). Cereal weed-competitiveness has been found associated with various morpho-physiological traits, such as early vigor, leaf area index, number of tillers, canopy architecture, and canopy height. Breeding new cultivars and genetic manipulation for improved weed-competitiveness requires quantification of these traits and elucidating their underlying genetic architecture. In that respect, the genomics revolution over the last decades resulted in highly mechanized genotyping methods with low costs that offer a vast amount of data in a short time [6]. Nevertheless, minor improvement in the plant trait data collection methods has been witness, especially for single plots or plants under field conditions [7]. This so-called phenotyping process is considered today as the main bottleneck of any breeding project [8]. It is usually performed by manual sample collection, which is labor-intensive, time-consuming, and subject to human bias. Furthermore, longitudinal growth analysis requires reputed evaluations, which cannot be performed using destructive measurements. Thus, there is an urgent need to develop novel high-throughput phenotyping (HTP) tools to overcome this phenotypic bottleneck and facilitate cereal breeding programs to enhance early vigor and weed-competitiveness [9,10].

Proximal and remote sensing methodologies are changing the field of plant phenomics and playing a major role in the detection of various developmental stages, responses to environmental cues and management, and their complex interactions (reviewed by [11–13]). A large number of the sensor are available for plant HTP including conventional off-the-shelf RGB cameras, multispectral or hyperspectral cameras, fluorescence sensors, optoelectronic sensors, LiDAR, or ultrasonic devices [14,15]. Generally, RGB-based HTP methods (i.e., image-based) are inexpensive, end user friendly, and do not require long or complex data-analysis methodologies, thus considered the most widely used tool [16]. Morphological analysis using image-based methods can be classified into two main approaches, which are oriented for the extraction of 2-D or 3-D morphological features. The 2-D approaches are based on the use of a single image but are limited to scarce morphological features such as width and cover area [17,18]. These parameters can be evaluated from close and remote platforms, thus, relevant for various growth scenarios and HTP tasks [14]. In terms of cereal breeding, Bacher et al. [19] evaluated a set of wild wheat introgression lines for the projected shoot area, PSA, (as an indicator of biomass) in response to water stress and were able to quantify the longitudinal responsiveness dynamics. This 2-D analysis approach offered the required sensitivity to identify promising lines, which exhibited high phenotypic plasticity during water stress. Torres-Sánchez et al. [20] employed a remote platform for accurate phenotypic evaluation of wheat green leaf cover area using an RGB camera mounted on an unmanned aerial vehicle (UAV). They achieved 92% evaluation accuracy, which remained robust over the entire area of the wheat field and under various imaging timings.

Stereovision-based models offer alternative plant HTP approaches based on 3-D morphological properties. Three-dimensional plant reconstruction has become a viable task with the advances in modeling algorithms and increased computational power, enabling the generation of image-driven models with higher resolution and a better description of plant canopy architecture [21]. Furthermore, 3-D reconstruction techniques allow a more comprehensive extraction of morphological parameters from a single organ to the whole plant level [22]. Such detailed analysis of plant morphology can be employed for capturing the dynamic growth and responses to the environment and to facilitate research and breeding efforts [23]. In that respect, Duan et al. [24] used the multi-view stereo (MVS) methodology to compare and evaluate early vigor trials between two wheat genotypes. The wheat plant height, tiller number, leaf number, and length were extracted by reconstruction of the plant's 3D shape. This approach was shown to be very efficient in differentiating between genotypes and demonstrate its potential contribution to cereal HTP. Despite the advantages inherent in these image-based HTP methodologies, the feasibility of using 2-D and 3-D image-driven models to evaluate and quantify cereal's competitiveness ability with weeds have not been tested nor validated under field conditions.

Furthermore, the contribution of different morphological parameters, and their potential to promote cereals weed-competitiveness ability is mostly unexplored.

Triticale (*Triticosecale* Wittmack; a hybrid of wheat (*Triticum* sp. ♀) × rye (*Secale* sp. ♂)), an alternative grain and forage cereal-crop was used here as a model for image-based phenomics of early vigor and weed-competitiveness in cereals. Two triticale lines with contrasting early vigor behavior were selected for the current study and were characterized via integrated sensing approaches with the aim to (**i**) evaluate the potential of using a close-range 2-D and 3-D modeling approaches as HTP tool for cereals early vigor traits under controlled conditions, (**ii**) identify and characterize key morphological traits that associate with weed-competitiveness, and (**iii**) validate the UAV-based 2-D and 3-D sensing approaches for early vigor detection under field conditions. The obtain results presents a comprehensive comparison between 2-D and 3-D HTP approaches, acquired from close and remote platforms, to detect early vigor related traits and facilitate high-throughput screening tools in cereal breeding for enhanced weed competitiveness.

## 2. Materials and Methods

### 2.1. Plant Material and Growth Conditions

Two triticale genotypes with different early vigor characteristics were selected for the current study: Triticale1 has tested in Israel two decades ago as forage crops and characterized by low initial growth, and X-1010 (PI 656390, NSGC) originating from Oregon (USA) and characterized by high early vigor and high stature. The model weed species used for competition assays were domesticated ryegrass (*Lolium multiflorum*) which is considered highly problematic in many areas of the world. The controlled conditions experiments were held in net-house under Mediterranean winter conditions. Plants were watered by an automated mini-sprinkler irrigation system (VibroNet™ 25 L/H, Netafim, Israel) as needed, and fertilized every two weeks with N:P:K (20:20:20).

### 2.2. Comparison of the Efficiency of Close-Range 2-D and 3-D Models to Distinguish between Genotypes under Net-House Conditions

The objective of this experiment was to compare plant morphological traits between the two genotypes using 2-D and 3-D models throughout the early developmental stages. For that purpose, five X-1010 and triticale1 seeds were placed in 1-L pots filled with Newe Yaar soil (57% clay, 23% silt, and 20% sand, on a dry-weight basis and 2% organic matter) and grown in a net-house. Seedlings were thinned to one plant per pot 7 d after sowing. A complete randomized design was applied, with six replicates (total 12 pots). Plants were imaged weekly seven times starting 21 d after sowing using a Nikon D7200 camera fitted with a Nikkor 18–140 mm lens. Images were acquired in the net house at approximately midnight under natural illumination.

The plant morphology was analyzed using two modeling approaches, 2-D and 3-D. For the 2-D analysis, plants were image as described previously [18]. One image was taken (each imaging session) one meter above the canopy from a nearly horizontal angle (Figure 1A), and the PSA was estimated using absolute units (cm$^2$) (Figure 1C).

For the 3-D analysis, 40 images were captured (each imaging session), irrespective of the developmental stage, in a hemispherical coverage, from a ~1 m distance from the plant canopy (Figure 2A). The overlapping area between consecutive images was approximately 75%. A structure from motion (SfM) multi-view stereo (MVS) using Pix4DMapper photogrammetric software (Version 1.4, PIX4d, Lausanne, Switzerland) was used for generating 3-D point clouds of the plants (Figure 2B). In general, the photogrammetric process involves the detection and matching of corresponding image key points and estimation of the image orientations. The orientation stage is based on a bundle adjustment model that simultaneously computes the image orientation and estimates the key point positions in 3-D. Because the reconstructed point clouds cover the entire scene, prior to the derivation and evaluation of the morphological parameters, a pre-processing stage was applied, which consisted

of scaling and cropping the plant-related points using meshlab software (Visual Computing Lab ISTI-CNR, Pisa, Italy) (Figure 2C).

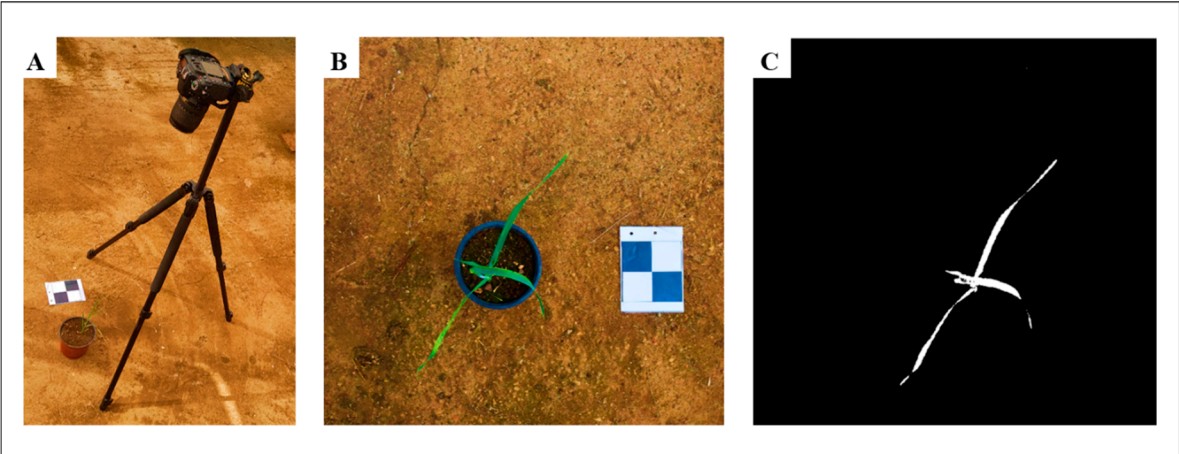

**Figure 1.** (**A**) Imaging setup of the 2-D phenotyping method. (**B**) A blowup image of the triticale plant and (**C**) the extracted projected shoot area (PSA).

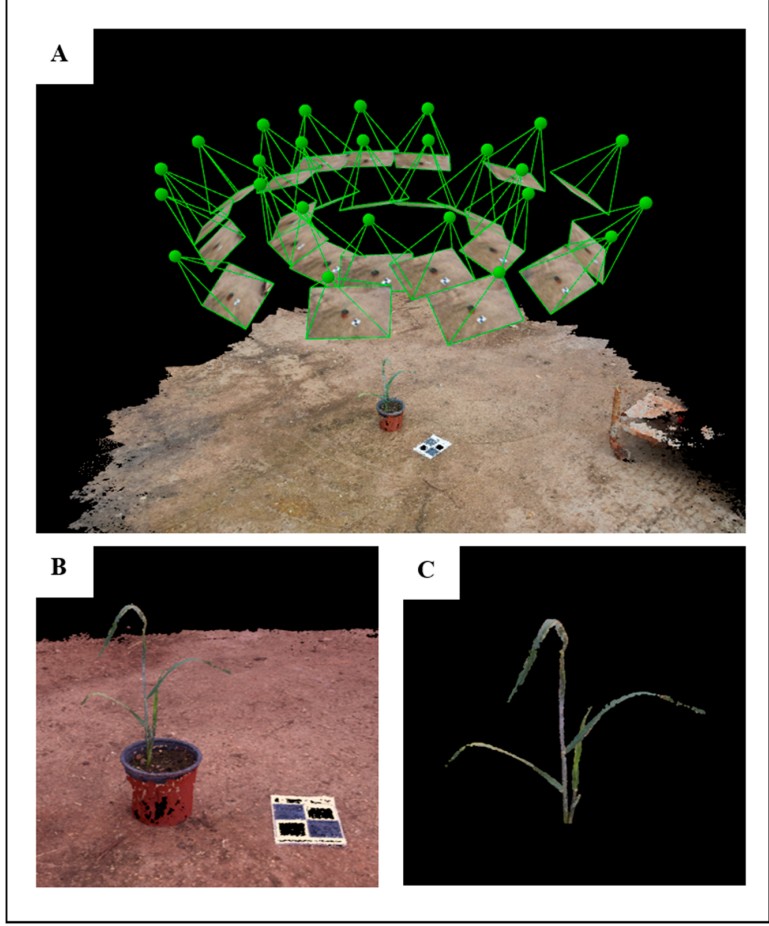

**Figure 2.** Main stages of the 3-D model reconstruction. (**A**) 3-D model of the entire net-house scene where the triticale plants were imaged, and the camera positions (green dots) orientations and imaged areas. (**B**) Blowup image of the 3-D model of a single triticale plant before and (**C**) after the pre-processing stage.

The point cloud was then exported as a text file with three columns (X, Y, Z) to MATLAB (R2017a, Mathworks® Massachusetts, USA) for the parameter extraction (Figure 3A). Four growth parameters were extracted from the 3-D model: volume, height, upper-width, and shoot convex area (SCA). Volume was estimated by the convex hull method, which fits a boundary around the entire x, y, z point cloud, and calculates the volume for that region (Figure 3B). Height was defined as the distance between the minimal and maximal z ordinates (Figure 3C). The upper-width was defined as the distance between the minimal and maximal x ordinates of the upper 10 cm points of the model (Figure 3D). For SCA, the convex approach was applied at the x, y ordinates that provide the projection of the shoot circumferential area onto a two-dimensional plane (Figure 3E).

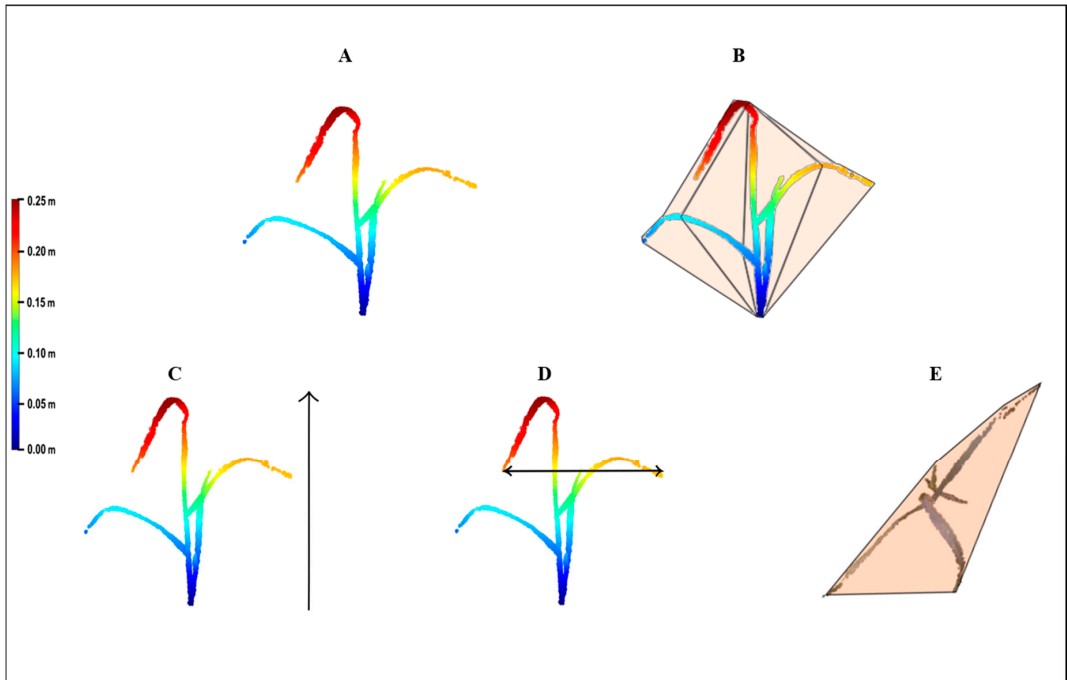

**Figure 3.** Demonstration of the parameter extraction using height map of the (**A**) 3-D model that included (**B**) volume, (**C**) height, (**D**) upper-width, and (**E**) shoot convex area (SCA).

### 2.3. Net-House Competition Experiment

The objective of this experiment was to identify the weed-competitiveness ability under controlled conditions between two triticale genotypes. The two triticale genotypes were sown in 16 L plastic containers ($32 \times 38 \times 20$ cm) with a mixture of 80% sand and 20% peat. Seeds were sown at a density of 250 plants m$^2$ that converted to three rows with eight plants. Seeds of ryegrass were sown together with the triticale at two infiltration rates: low (30 plants m$^2$) and high (96 plants m$^2$). Triticale and weed development evaluations were performed 6 and 12 weeks after sowing. The above-ground biomass of the weed was harvested, and their dry weight was determined. Then, triticale PSA was evaluated as previously described, followed by shoots dry weight determination. For the dry weight evaluation, samples were oven-dried at 70 °C for 48 h and weighed. The experiment was conducted in a complete randomization design with four replicates.

### 2.4. Comparison of the Efficiency of UAV-Based 2-D and 3-D Models to Distinguish between Genotypes under Field Condition

The objective of this experiment was to validate under field conditions the ability of the image-based HTP methods to identify differences in morphological features between the two triticale genotypes and to link them to weed-competitiveness ability. Experiments were conducted during the winter of 2019–2020 at a rain-fed field at Menashe Plain, 32.602 N 35.0512 E, with an annual rainfall of 530 mm.

The soil type at the site is Rendzina with a 5% slope. Plots were sown on 24 November 2019 by a Dönder No-till seeder. The plot size was 4-m-wide by 10-m-long, with genotypes sown in complete random block design with six replicates. The seeding rate was adjusted for 220 plant m² commercial field stand. Weed infestation was based on the natural seed bank. Fertilization treatments were similar to those delivered to the adjacent commercial field. Full emergence in all plots was recorded on 15 December 2019, a week following the first significant shower. Two months after sowing, shoot weeds dry weight was documented from 0.5 m² sections collected randomly from the center of each plot. Then, the plots were imaged using UAV for triticale growth evaluation.

Flights conducted 30 m above ground level, resulting in ground sampling resolution of ~0.6 cm/pixel, using a DJI Phantom 4 Pro UAV (Shenzhen, Guangdong, China). The UAV is equipped with a 20 M Pixel RGB camera with wavelengths at 620–750 nm (R, red), 495–570 nm (G, green), and 450–495 (B, blue). Mission planning was conducted using DJI Ground Station Pro (GSP) software. All the technical specifications of the UAV are listed in Table S1. The flight was designed with 80% image overlap along flight corridors. Then, Pix4DMapperPro desktop software (Pix4D SA, Switzerland, http://pix4d.com) was used to generate orthomosaics by implementing a structure-from-motion (SFM) algorithm (Figure 4A,B). Six ground control points (GCP) geolocated with Real-Time Kinematic (RTK) survey precision were used to georeference the orthomosaics (Figure 4A). The Pix4D processing options were done according to Pix4D's "3D Maps" template version 4.1.10.

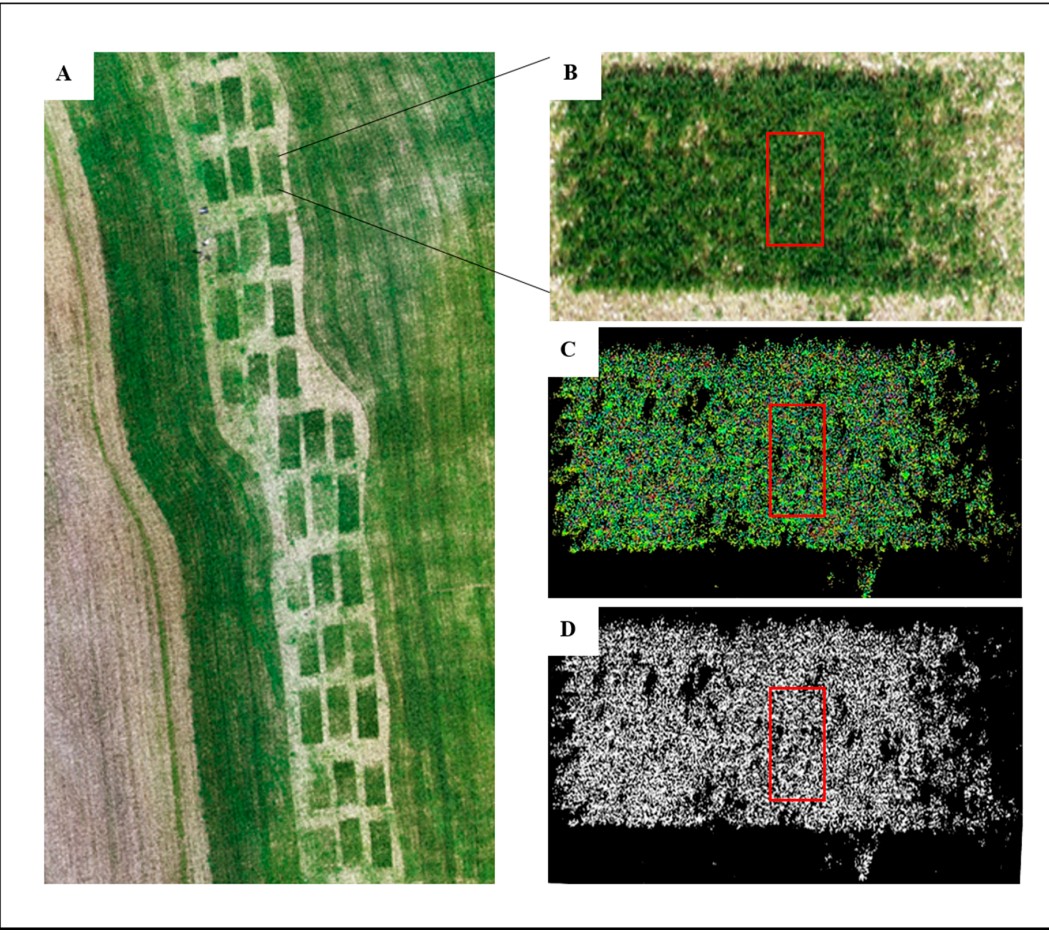

**Figure 4.** Main phenotyping stages extracted from the UAV in the field experiment. (**A**,**B**) Stitching the originally acquired images to orthomosaics and placing the treatment plots polygons respectively. (**C**) Extraction of the plot average ExG value and (**D**) projected shoot area (PSA) following the soil/crop classification.

## 2.5. 2-D Parameter Extraction by Remote Sensing

Plant development was found to have a close relationship with various growth indices (Das et al. 2016). Thus, the Excessive Green (ExG) values of the triticale plots were extracted as an indication for their development (Figure 4C). The ExG values were derived from the georeferenced orthomosaic GeoTIFFs that were generated from the UAV flights using the equation suggested by Woebbecke et al. [25]:

$$ExG = 2 \times G - R - B \tag{1}$$

Plot-level ExG means from UAV's were created in ArcGIS® 10.6 (ESRI, Redlands, CA, USA). Shapefiles containing annotated single plot polygons were generated. ExG plot means were generated using the Zonal Statistics as table function in ArcGIS (Figure 4B). Following the ExG evaluations, PSA evaluation of the triticale plots was held by applying a threshold value to the ExG monochromatic orthomosaic, resulted in a binary image, where white pixels represent vegetation, and the black pixels represent the surrounding soil (Figure 4D). Here, the Outsu threshold was used [26]. The Outsu algorithm provides a single intensity threshold that classifies pixels into two groups, soil, and vegetation in this case, by minimizing the variance of the intra-class intensity values [27]. Classification and segmentation of the vegetation pixels allowed the evaluation of triticale pixels exclusively for the PSA evaluations (Figure 4D). This procedure was implemented using an image analysis package in Arcmap.

## 2.6. Parameter Extraction by 3-D Remote Sensing

The 3-D point clouds were generated from the sets of UAV images using Structure from Motion (SfM) algorithms by Pix4DMapper software. The UAV imagery were collected on two flights. The first flight collected data immediately after planting over bare soil with no vegetation on 24 November 2019. The photogrammetry-based point clouds of the vegetation-free plot were used for generating the Digital Elevation Model (DEM) (Figure 5B). The second flight collected data over the emerged triticale plants, on 5 February 2020. The respective point clouds were used for generating the Digital Surface Model (DSM) generation (Figure 5A). The point clouds of each flight were converted to raster layers using the Inverse Distance Weighting (IDW) interpolation method with a spatial resolution of 1 cm using the Pix4DMapper software (Figure 5C,D). A canopy height model (CHM) for each sampling day was generated by computing the pixel-wise difference between DSM and DEM, respectively [28]. Mean canopy height were extracted for each sampling plot from CHM using zonal statistical in Arcmap 10.6 (Figure 5E). The mean canopy volume of each sampling pot was calculated by multiplication between mean canopy height and cover canopy area.

## 2.7. Statistical Analyses

The JMP® (Ver. 15) statistical package (SAS Institute Inc., Cary, NC, USA) was used for statistical analyses. Each growth parameter extracted from the X-1010 and Triticale1 was compared using a *t*-test at a 5% level. These included the 2-D (PSA), 3-D (height, volume, upper-width, and SCA), and biomass (weeds and triticale) from the net house trial, along with the PSA height, volume, and weed biomass parameters from the field trial. Then, for data extracted in the first net house experiment, nonlinear regressions were held using Sigmaplot 14.0 software (SPSS Inc., Chicago, IL, USA). The morphological traits of each parameter were nonlinearly regressed with time using three parameters sigmoid equation:

$$Y = \frac{a}{1 + e^{\frac{-(x-x0)}{b}}} \tag{2}$$

When $Y$ is the value of the analyzed trait, $x$ expresses time (days after sowing), $a$ is the maximum value of the trait, $x_0$ is the time value required to accumulate 50% of the morphological trait, and $b$ is the slope at the inflection point [29]. The extracted parameters from each trait were compared between the genotypes. For the field trial data, a linear model was fitted between the weed biomass

and each parameter that was extracted from the UAV. Root square mean error (RMSE) and coefficient of determination ($R^2$) were computed for model comparison.

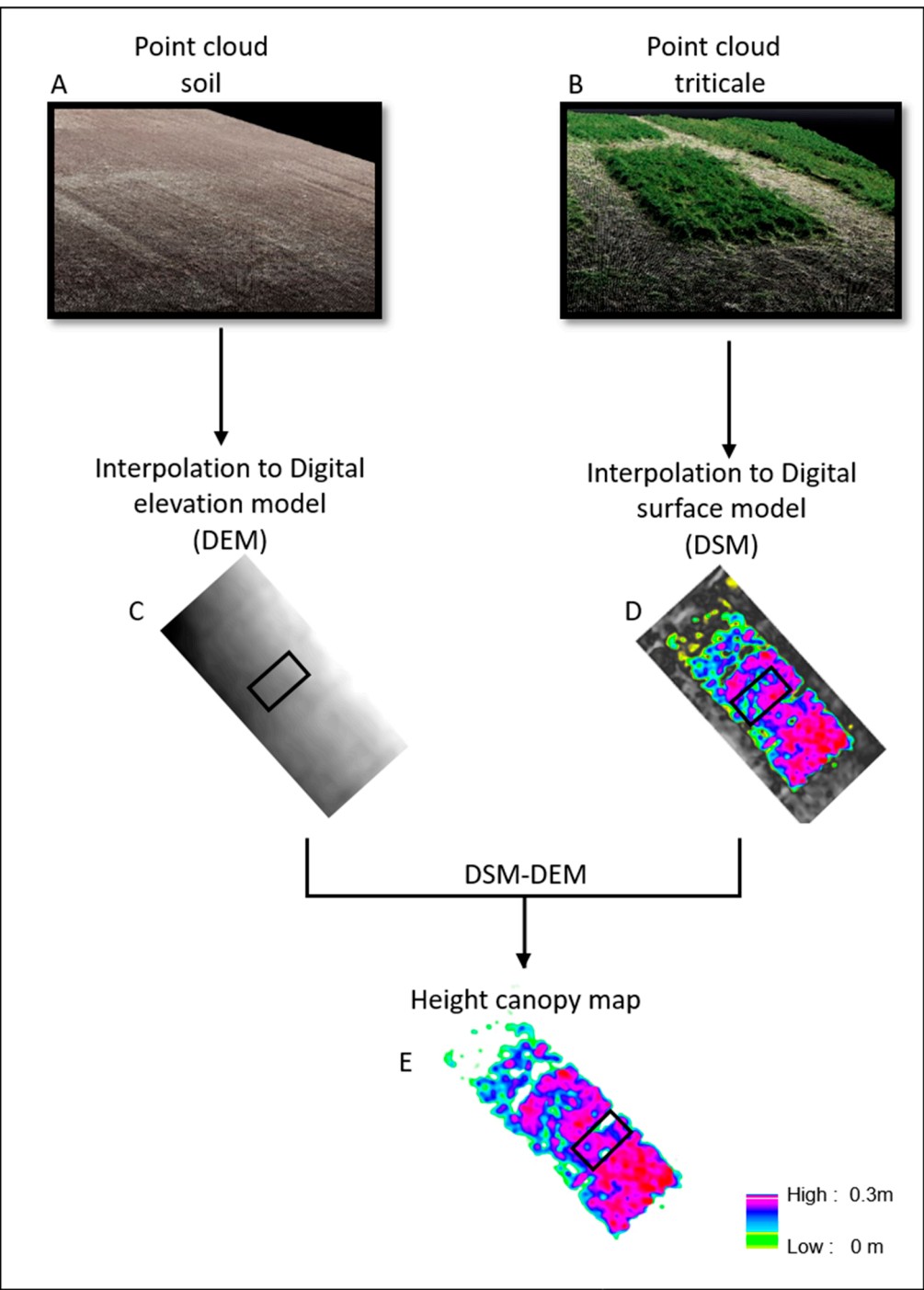

**Figure 5.** A workflow diagram of the 3-D canopy morphological parameters extracted for the triticale plant in the field trial using the UAV images. 3-D point cloud of the experimental plots (**A**) before and (**B**) after the triticale emergence and development, the corresponding (**C**) Digital Elevation Model (DEM) and (**D**) Digital Surface Model (DSM), and (**E**) their subtraction result, canopy height model.

## 3. Results

### 3.1. Early Growth Dynamics of Single Plant Can be Monitored Using Close-Range 2-D and 3-D Models

To test the accuracy, reliability, and earliest detection timing abilities of the 2-D and 3-D modeling approaches to capture differences in the values and the longitudinal dynamics of morphological traits associated with early vigor, two triticale genotypes with contrasting vigor were used. The 2-D model enables calculating a single trait of PSA, and the 3-D model provided four morphological traits (height, volume, upper width, and SCA). In general, both modeling approaches reflected the early growth rate superiority of X-1010 compared to Triticale1 during the initial growth stage of 56 DAS. Furthermore, the *t*-test revealed that the differences in the growth parameters of the two genotypes were significant as early as 21 DAS for all parameters suggesting the equal potential of all parameters to reflect early vigor ability (Figure 6). However, each parameter showed different consistency over time and different timing with a significant and highest difference. Height was the only parameter that demonstrated significant differences between the values extracted from X-1010 and Triticale1 over a consecutive four weeks period between 21 and 42 DAS (Figure 6). Shoot convex area was the weakest indicator with single timing (21 DAS) sowing significant differences between the genotypes. In terms of the magnitude of the differences, 21 DAS volumes generated the highest difference between the genotypes (65%), and SCA was the second-best parameter with a 50% difference. Two weeks later (35 DAS), upper-width was the parameter generating the highest different (77%) between the genotypes, while the SCA values were not significantly different between X-1010 and triticale1 genotype (Figure 6). The 2-D parameter, PSA, had significantly higher values in the X-1010 compared to the triticale1 between 21 and 49 DAS, with exceptional at 35 DAS. The highest difference in this parameter was observed at 42 DAS, 45% (Figure 7).

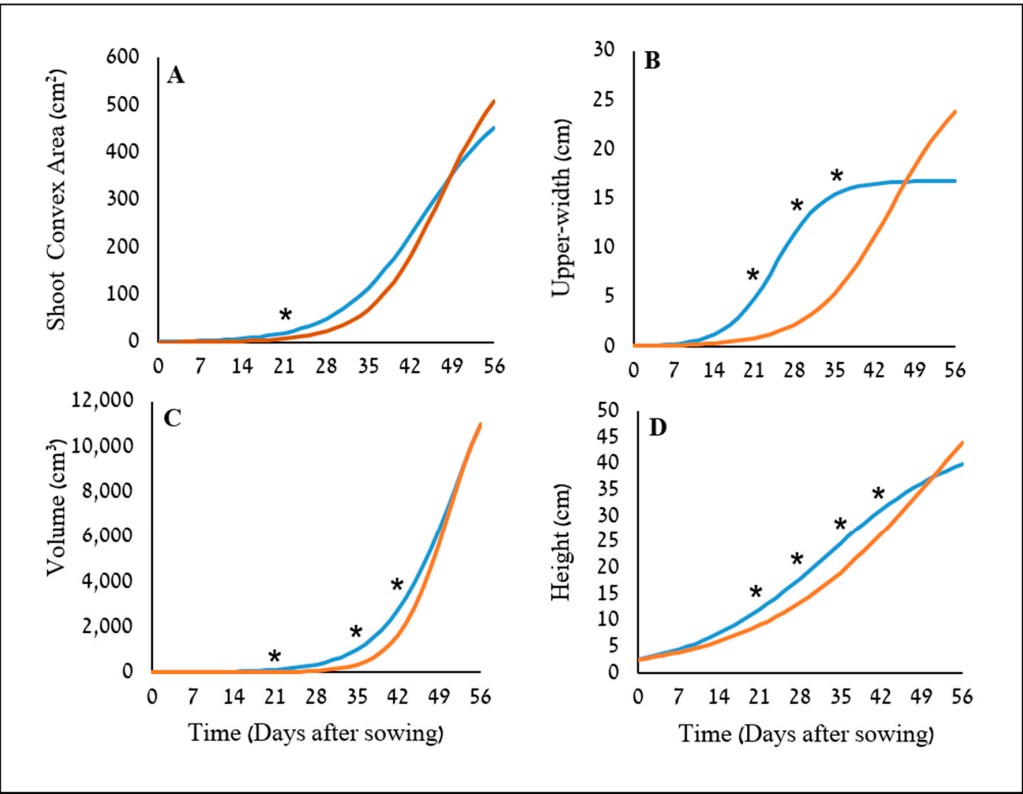

**Figure 6.** Sigmoid relationship between time (days after sowing) and X-1010 (**blue**) and Triticale1 (**orange**) genotypes (**A**) shoot convex area (SCA), (**B**) upper-width, (**C**) volume, and (**D**) height values that were extracted from the 3-D model. * indicate a significantly difference between X-1010 and Triticale1 ($p \leq 0.05$) according to *t*-test ($n = 6$).

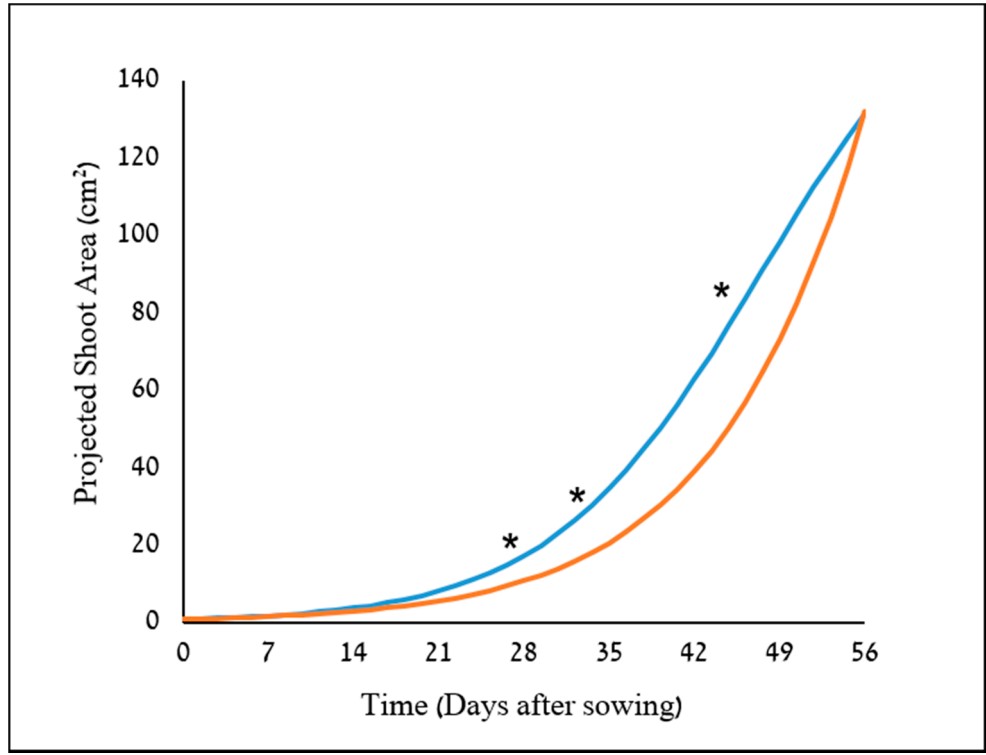

**Figure 7.** Sigmoid relationship between time (days after sowing) and X-1010 (**blue**) and triticale1 (**orange**) genotypes projected shoot area (PSA) values that were extracted from the 2-D model. Asterisks indicate a significant difference between X-1010 and Triticale1 ($p \leq 0.05$) according to the *t*-test ($n = 6$).

The nonlinear sigmoid analysis aimed to provide deeper insight, into the dynamic aspect of the morphological changes between the genotypes and to emphasize the parameter with the greatest difference between the genotypes. Here, the $X_0$ value was compared to reveal variations in the growth rate pattern. For the 3-D modeling approach, the upper-width was the most significant growth parameter with a 46% difference in the $X_0$ value, 24.5, and 44.8 days for the X-1010 and the Triticale1, respectively (Table S2). The width-related $X_0$ value was the lowest compared to values extracted from other growth parameters, e.g., 32 and 52 days for the height and volume, respectively, suggesting early indication ability for early growth traits associate with upper-width. The $X_0$ value of plant volume, which was highly significant in the *t*-test analysis, had only a 4% difference between X-1010 and triticale1, 52.5 and 50.4 days, respectively (Table S2). For the 2-D parameter (PSA), 38% differences in the $X_0$ values were observed between the X-1010 and triticale1, 47 and 74 days, respectively (Table S2).

### 3.2. Early Vigor Affects Weed-Competition Ability

Weed-competition ability differences between the triticale genotypes at their early growth stages were examined under two weed-density conditions. The triticale and weeds development was determined using dry weight accumulation. Figure S1 provides a qualitative indication for the X-1010 higher early vigor, and the resulted improved weed-competitiveness ability compared to the Triticale1. The image that was taken 6 WAS, shows the better establishment of the X-1010 plants compared to the triticale1 genotype, and higher leaf area coverage can be clearly observed (Figure S1A). Correspondingly, the ryegrass plants that grew next to the X-1010 plants were significantly stunted in terms of phenology stage, size, and biomass accumulation, compared to plants that grew next to the triticale1 genotype (Figure S1B).

Similarly, the PSA value extracted from the 2-D model quantitative reflected the significant improvement in the early growth ability of the X-1010 genotype compared to the Triticale1. Extracted PSA values were significantly higher ($p \leq 0.025$) in the X-1010 at both evaluation timings

(6 and 12 DAS) and under both weed densities, suggesting the robustness of PSA as an early-growth indicative parameter. For example, the median PSA values recorded 6 WAS under the high weed density level were 190% and 125% of weed-free control for the X-1010 and Triticale1 genotype, respectively (Figure 8A,B). The higher PSA values of this genotype resulted in significantly ($p \leq 0.043$) lower weed biomass under all scenarios, indicating an improved ability to compete with weeds. The weed dry weight median value (6 WAS under the high density) was lower by 60% in the X-1010 compared to the Triticale1 (Figure 8C,D). In terms of biomass, non-significant differences were observed between the two-triticale genotypes, except in 12 WAS with low weed density. Furthermore, under high weed density, the computed $p$ values ($\leq 0.4491$) in the comparison between the genotypes suggest similar biomass accumulation (Figure 8E,F).

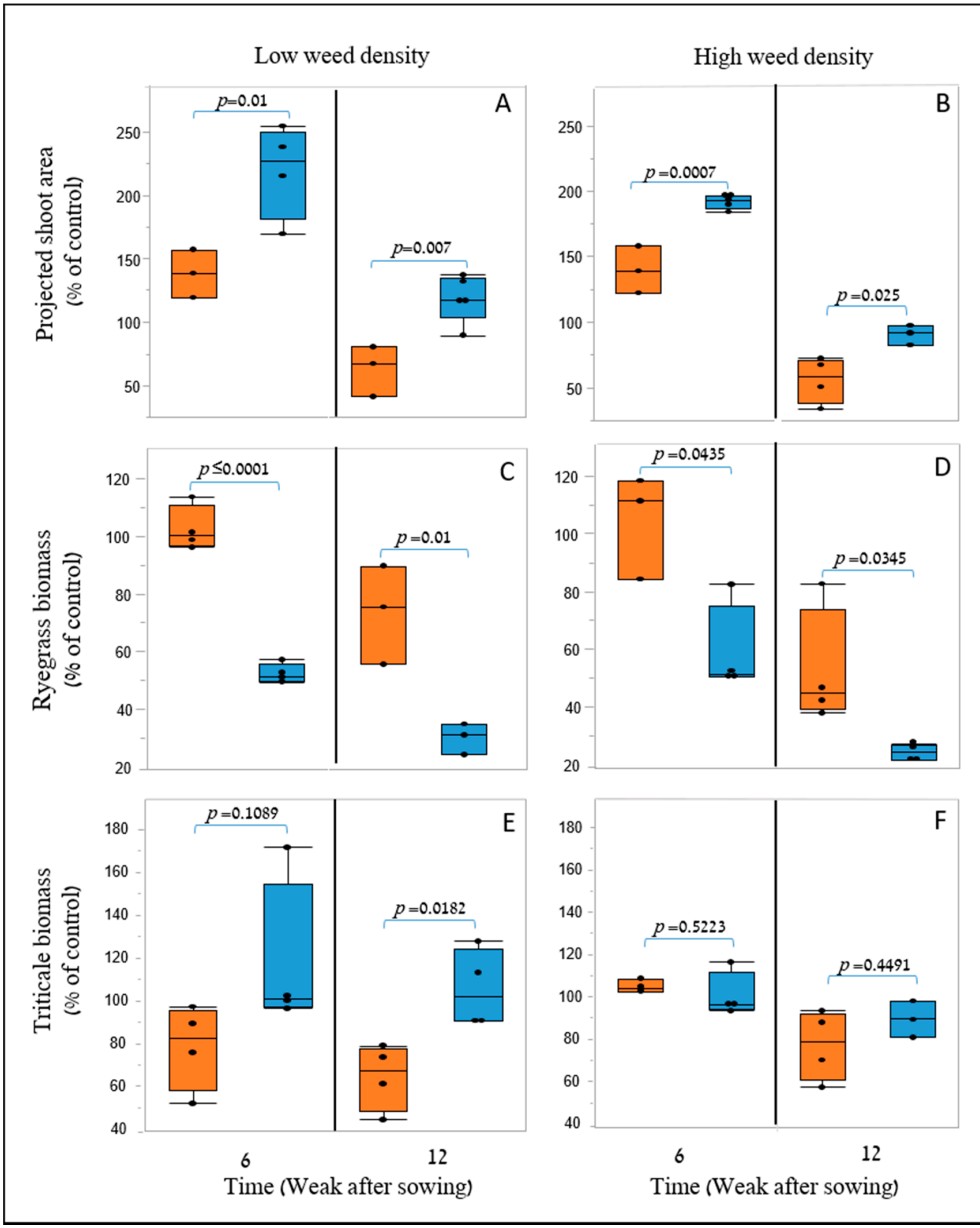

**Figure 8.** Results of the controlled conditions competition experiment under low (left, **A**,**C**,**E**) and high (right, **B**,**D**,**F**) *Lolium multiflorum* (ryegrass) density levels. (**A**,**B**) X-1010 (**blue**) and Triticale1 (**orange**) projected shoot area, (**C**,**D**) ryegrass shoot biomass, and (**E**,**F**) triticale genotypes shoot biomass $p$-values were determined by $t$-test ($n = 6$) between the X-1010 and the Triticale1 genotype.

### 3.3. UAV-Based Traits Provide Indication for Early Growth under Field Conditions

The field experiment aimed to evaluate and validate the ability of the 2-D and 3-D morphological traits to monitor differences in early growth between the triticale genotypes under real field conditions and to assess the contribution of each parameter for the competitiveness ability. Here, growth parameters were extracted from UAV and included PSA, height, and volume. All traits were significantly higher for the X-1010 compared to the Triticale1 genotype indicating the ability of these parameters to detect faster establishment at the early growth stages. The 2-D parameter, PSA, median value for the X-1010 was more than four-fold higher than the Triticale1 genotype, 0.22 and 0.05 cm$^2$, respectively (Figure 9A). The 3-D parameters, height, and volume were also significantly ($p = 0.0001$ and $p = 0.023$, respectively) higher in the X1010 compared to the Triticale1, which further indicates the better establishment of that genotype (Figure 9B,C). However, in the case of height, only 1.5 fold increase was observed between the median values of X1010 and Triticale1. Correspondingly to the morphological differences between X1010 and Triticale1, the weed biomass in the X-1010 plots was significantly ($p = 0.014$) lower compared to Triticale1 (Figure 9D). When the extracted morphological parameters were linearly regressed against the weed biomass, the PSA model resulted in the highest R$^2$ (0.5) and lowest RMSE (6.2 g) values suggesting the impact of the PSA on the weed-competitiveness ability of the triticale genotypes (Table 1).

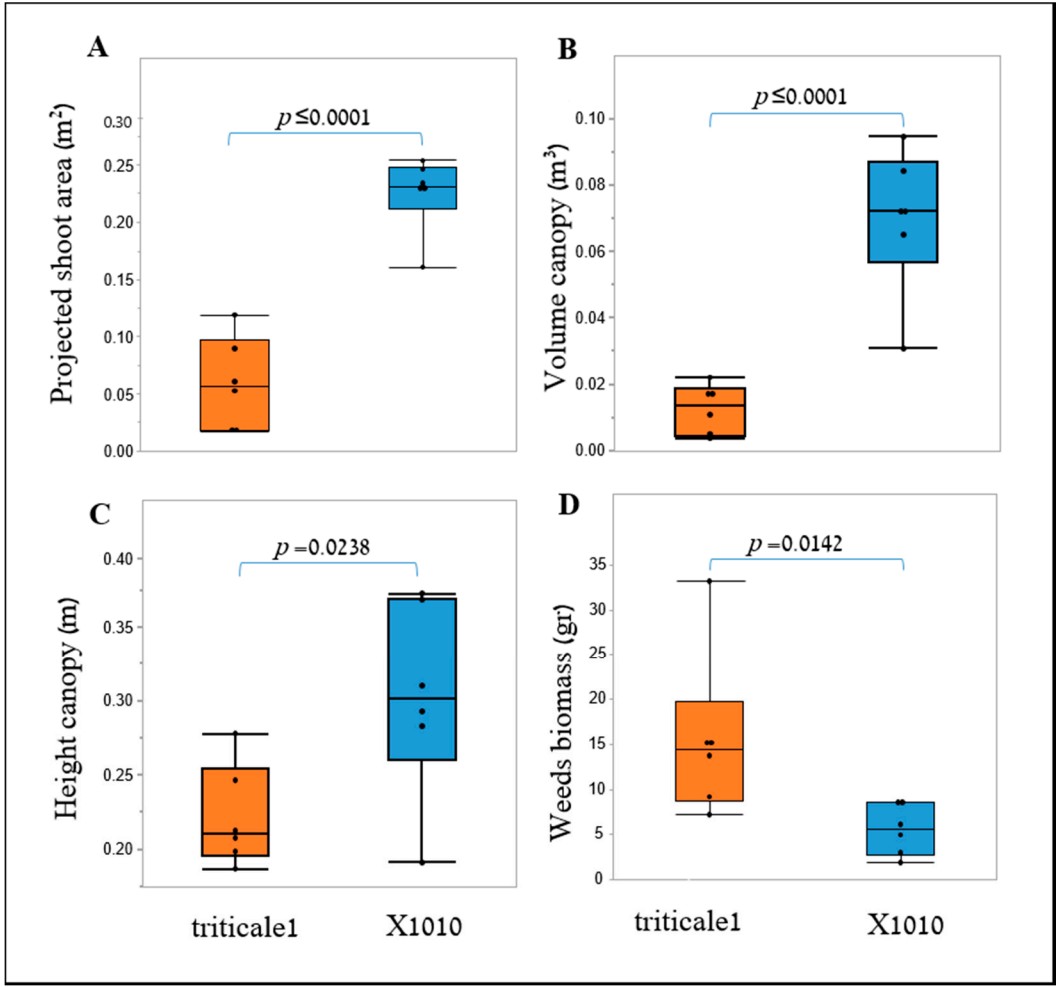

**Figure 9.** Results of the field conditions competition experiment. Compression between X-1010 (**blue**) and Triticale1 (**orange**) genotype (**A**) projected shoot area (PSA), (**B**) volume, (**C**) height, and (**D**) resulted in weed biomass. *p*-values were determined by *t*-test (*n* = 6) between the X-1010 and the Triticale1 genotype.

**Table 1.** Coefficient of determination ($R^2$) and root mean square error (RMSE) values from the linear regressions between weed biomass and triticale projected shoot area (PSA), height, and volume values that were extracted by the UAV in the field experiment.

| Growth Parameter | $R^2$ | RMSE |
|:---:|:---:|:---:|
| PSA | 0.50 | 6.2 |
| Height | 0.16 | 6.8 |
| Volume | 0.39 | 8.0 |

## 4. Discussion

Cultivars with enhanced weed-competitiveness, i.e., fast early canopy growth rate, can improve the adoption of integrated weed management practices in cereals, reduce herbicide use, and promote sustainable agro-system production [30–33]. Currently, phenotypic selection methodologies for competitiveness-related trait breeding in cereal involve long and tedious, and in many cases destructive, tools [34]. Advanced HTP tools that provide automated and non-biased growth evaluations in a non-destructive manner offer a new toolkit for breeding that can minimize the phenotyping bottleneck [34]. The potential use of two image-driven modeling approaches (2-D and 3-D) to extract and evaluate morphological features of cereal plants at their early growth stages were demonstrated, which in return, provided an indication for their weed-competitiveness ability. Two triticale genotypes were used as model cereals in this study, X-1010 (high early vigor), and Triticale1 (low early vigor). The morphological advantage of X-1010 was observed only at the initial growth stages (Figures 6 and 7), while at later stages, the Triticale1 genotype growth rate becomes similar or higher. This trend of growth indicates that the two triticale genotypes used in this study were suitable for comparative early growth assay, as demonstrated elsewhere [35]. Furthermore, weed biomass was lower when grown next to the X-1010 at the field experiments. These results indicate that the differences between morphological parameters of these two genotypes detected by the suggested HTP tools at early growth stages were actually translated into weed growth suppression (Figures 8 and 9).

Two plant-shape modeling approaches, 2-D and 3-D, were applied. The two approaches were tested from a close-range (handheld consumer-grade camera) and remote (UAV) platform. From both ranges, the two approaches were sensitive and identified differences in the genotypes morphological parameters, and therefore are suitable for early vigor selection. For the single plant close range phenotyping, PSA values showed a significant difference between the two triticale genotypes starting 21 DAS with only one exertional at 35 DAS. However, plant upper-width and height were even more sensitive parameters that showed the greatest differences between the genotypes through the net-house study, in terms of the magnitude of differences and repeatedly, respectively. The dynamic analysis using the sigmoid relationship revealed that the upper-width also provides the lowest $X_0$ value for the X-1010 with the greatest difference from the corresponding $X_0$ value of the Triticale1 genotype. This means that even when the morphological parameter is analyzed on a large time scale (not at specific timing), plant upper-width remains a favorable indicator for early growth. Upper-width may not be a favorable parameter for field experiments with overlapping crop-plants canopies. However, many breeding and genetic studies are conducted under controlled conditions using individual plants [19] and may find this parameter relevant and of great interest. Height showed the greatest consistency in demonstrating the advantage of the X-1010 over the Triticale1, suggesting its credibility for indicating early vigor.

In the field under the weed-competition conditions, PSA and height remained robust early-growth indicators and showed significant differences between X-1010 and the Triticale1 genotype. These results suggest the stability of these two parameters as early vigor indicators over varying conditions that mainly include the crop population-level (not just imaging of a single plant) when each plant experience intra-species competition with surrounding triticale plants. The significant differences in the extracted morphological parameters also demonstrate that the 2-D and 3-D modeling approaches can be used for early vigor detection when acquired from remote platforms such as UAV. Furthermore, the greater PSA and height values observed in the X-1010 were coupled with lower weed biomass, which suggests

that these parameters are suitable proxies for cereal's weed-competition ability under field conditions. However, PSA showed the most significant linear correlation with weed biomass providing quantitative evidence for its greater impact on the growth of the surrounding weed compared to height or volume. These results demonstrate the advantage of PSA over height for weed competition prediction and as an effective parameter for the screening process [36]. They also emphasize that the weed-competitiveness trait is affected by plant and leaf morphology and not exclusively by the vertical growth accumulation. Despite their similarity, PSA was favorable over SCA (extracted by the 3-D model) as it better reflected the actual development of the plant and not just the permitted-like area that its incensement through time is limited.

Height and other canopy 3-D architect parameters were previously reported as indicative for early-vigor and weed competitiveness [37]. However, the reconstruction of the 3-D model, which is essential for these parameters extraction, involved with long processing time and high computation power, mainly in the close-range method that required 40 images per single plant [22,38]. The orientation and scaling stages are performed manually, thus, screening of a large number of genotypes might increase time costs [38]. Furthermore, the 3-D reconstruction software might not be operational-friendly for end-users, and to achieve accurate results in the longitudinal evaluations RTK-based geo-referencing is needed. This device may not be available in any biology and genetic oriented labs, thus, the 3-D modeling approach may not be suitable to apply in the breeding program, which prioritizes friendlier HTP tools [12,39]. In that respect, the 2-D modeling approach, which yields the PSA parameter, may be favorable over the 3-D one, mainly for the close-range models. A single image with short (~5 sec) processing is needed for a single plant, thus, can be relevant for repeated measurements of large-scale collections.

PSA was previously identified as an indicator for crop development in general [40], and particularly for cereal early vigor [41]. However, this study correlates PSA to weed biomass repression and demonstrates the contribution of PSA to weed-competitiveness ability and the linkage between these two parameters. In this study, PSA values were extracted from the UAV and handheld consumer-grade camera. Both methods are highly available and end-user friendly HTP tool. They are less pricey compared to other HTP tolls such as multi/hyperspectral camera or depth cameras [18], thus, highly available [42]. Close range PSA extraction can be achieved using online free software, which emphasizes the high practicability of this parameter [43]. UAVs offer field-oriented HTP tools that can be used on large scales. This HTP tool is suitable for the early stage screening of breeding programs where a large number of lines are scanned in small plots. For example, in the field experiment, 18 plots were imaged within 20 min. Accurate PSA estimations require weed-free plots, which were hand weeded in this experiment. In the future, advanced weed/crop classification algorithms can be used prior to the PSA estimation to remove the weed coverage (e.g., [44]), however, these models add complexity to the proposed methodology and were not within the scope of this study.

## 5. Conclusions and Future Perspective

Two HTP approaches for early vigor associated-traits selection in cereal were applied in this comparative study. Both approaches were successful in detecting differences between morphological features along the early growth stages of the two contrasting triticale genotypes, with PSA, height, and upper-width being the most indicative traits. However, PSA is based on a 2-D model, thus, it can be extracted more easily and characterized by the short processing time and more suitable to HTP. When PSA was extracted under weed infestation conditions, in the net-house, and the field experiments, it provided adequate indication for cereals weed-competitiveness ability and better correlation to weed biomass. PSA can be extracted from proximal and remote sensing platforms, and therefore is a useful parameter for controlled and field conditions studies. Consumer-grade cameras and UAVs can be used for PSA evaluation, which turns it to highly available and user-friendly for non-technological oriented labs or breeders. In this study, the different morphological parameters were analyzed separately to monitor early-vigor characteristics. However, advanced multivariate machine learning methods

(e.g., partial least square analysis (PLSR), Random Forest) can be used in the future on similar data sets to provide more robust and sensitive detection of early growth variation across genotypes. Results from this study will contribute to enhancing cereal breeding programs to improve early vigor and weed-competitiveness. In a wider perspective, it can also serve as a platform for HTP pipelines of other plant architecture trait-based breeding programs.

**Supplementary Materials:** The following are available online at http://www.mdpi.com/2072-4292/12/23/3877/s1. Figure S1: Illustration of the controlled conditions competition experiment. (**A**) X-1010 and (**B**) Triticale1 genotype plants imaged 6 weeks after seeding, and correspondingly, *Lolium multiflorum* plants that developed next to (**C**) X-1010 and (**D**) Triticale1 genotypes, Table S1: Technical specifications of the DJI Phantom 4 Pro UAV used in the field experiment., Table S2:Coefficient and their standard error (in prentices) estimates from the sigmoid regression between X-1010 and triticale1 genotype volume (cm$^3$), shoot convex area (SCA) (cm$^2$), height (cm), upper-width (cm) and projected shoot area (PSA) (cm$^2$) with time (day), and their respective Root Mean Square Error (RMSE) and coefficient of determination (R$^2$). Values were extracted were made between 7 and 56 days after sowing. Leaf projected shoot area was extracted from the 2-D model while height, volume, upper-width and CA from the 3-D model.

**Author Contributions:** Conceptualization, Z.P., R.N.L. and R.B.-D.; Funding acquisition, Z.P., R.N.L., E.A. and R.B.-D.; Investigation, S.A.; Methodology, Z.P., R.N.L., E.M and R.B.-D.; Supervision, R.N.L. and Z.P. All authors have read and agreed to the published version of the manuscript.

**Funding:** This study was supported by the Chief Scientist of the Israeli Ministry of Agriculture (grants # 12-01-0005, 837-0134-13, and 44-03-0002).

**Acknowledgments:** We thank members of the Peleg, Ben-David, Argaman, and Lati labs for their invaluable assistance in the field experiments.

**Conflicts of Interest:** The author declares no conflict of interest.

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
