# Peer review of "Image-Based High-Throughput Phenotyping of Cereals Early Vigor and Weed-Competitiveness Traits"

_remotesensing, doi:10.3390/rs12233877_

Round 1

Reviewer 1 Report

  1. Comments to Author:

Title: Image-based High-throughput Phenotyping of Cereals Early vigor and weed-competitiveness Traits.

1.1 Overview and general recommendation:

The study is on a topic of relevance and general interest to readers of the journal. The authors propose a method to monitor early vigor of crops under weed-competitiveness that enables the early identification of superior candidates. In the proposed methodology, the sensitivity of 2-D and 3D morphological parameters to detect early vigor is evaluated under both controlled and field conditions.

I found the manuscript to be well organized and written. However, I have little confidence in some parts and I cannot recommend it for publication as is stands. The concerns are explained in more details in the following paragraphs.

1.2 Major comments

The main concern about the manuscript is with respect to the methodology. The rapid transferability of morphological parameters between scales: single plant, controlled net-house, and field conditions overlooked important technical details.

Early statements about outcomes (abstract) of the use 2D and 3D parameters (abstract, line 30-32) to estimate early vigor were partially fulfilled further in the results section. Morphological parameters were not consistently used through different scales. 2-D, 3-D, and PSA parameters were utilized in single plant pots, PSA was only used in net-house, and PSA and EXG intensity used in field conditions. This added gaps of information, leaded to partial discussion and conclusions, and added confusion to reader. Not sure why 3-D parameters were not included in net-house and field conditions, UAV-based photogrammetry is widely used over large breeding nurseries and the price of software is not a strong argument.

The outcomes of the most importance 3-D and 2-D morphological parameters defined at single plants pots does not directly transfer to the canopy level, where plant architecture and morphological parameters can be affected when neighbor plants are competing for light and resources. The assumptions about the direct transfer of 2-D and 3-D parameters between single plant to canopy level deserves a brief informative reference for reader.

The use of excessive greenness index and thresholding are common methods used for detection of green and non-green objects in imagery and widely reported. Thresholding requires manual tuning, which significantly constrains scalability. In addition, the proposed method highly depends on efficient manual cleaning of weeds prior to UAV data collection. This point becomes mandatory for the pipeline to succeed by preventing misclassification of weeds as target crop, or it constraints the applicability only under idealistic low weed pressure conditions.

1.3 Minor comments

Abstract

Line 21. “The adequate phenotyping ability for early growth and competition was robust under controlled and field conditions.” But if I understood correct, the performance of the method is highly dependent on efficient manual cleaning before UAV data collection to ensure that PSA is not inflated by incorrectly including weeds as target crops.

Introduction

Line 97. “iii) validate these phenotyping approaches under field conditions. Our results present a comprehensive comparison between 2-D and 3-D phenotyping approaches to detect early vigor related traits and facilitate high-throughput screening tools in cereal breeding for enhanced weed competitiveness.” This statement was not fulfilled, since there was not comparison between 2-D and 3-D in net-house and field conditions, 2-D and 3-D comparison was only carried out in single plant pots, not competing for light and resources.

Materials and Methods

Line 205. “Classification and segmentation of the vegetation pixels allowed the evaluation of triticale pixels exclusively for the PSA evaluations (Figure 4D).” In field conditions, it assumes a previous very efficient manual cleaning of the plots, then all greens pixels assumed as triticale.

Results

Line 228. It is worthy to inform that results in 3.1 were based on single plant pots.

Line 260. “Here, the X50 value”. X50 concept was not previously defined in the text. I assume this is a typo and it refers to x0 in equation 2.

Line 300. “Here, growth parameters were extracted from UAV and included PSA and ExG index value, which reflects the triticale growth by coverage area and reflectance intensity”. EXG spectral intensity is now included to characterize early vigor, but its use was not described in materials and methods, and it was not used in net-house scale, which shows an inconsistent utilization of parameters throughout different scales.

Discussion

Line 337. X50 was not previously defined.

Line 348. “For these reasons, the 3-D modeling approach may not be suitable to apply in the breeding program, which prioritizes friendlier HTP tools [9,33].” UAV-based photogrammetry is widely used in large field nurseries in breeding programs. The price of the photogrammetry software is not a prior strong argument to discard 3-D parameters in net-house and field conditions in results and discussion sections. GUI Meshlab, OpenDroneMap are open source and free available alternatives to process ground or UAV-based RGB imagery.

Author Response

November 16th 2020

Dear Editor,

Remote Sensing

Re: Revision of “Image-based high-throughput phenotyping of cereals early vigor and weed-competitiveness traits (remotesensing-989904).

Thank you for your letter dated November 02, 2020, notifying us that our manuscript may be acceptable for publication in Remote sensing following major revision.

The manuscript was revised according to the comments made by the two reviewers (in Italics) as specified below.

In response to the comments of reviewer #1:

The main concern about the manuscript is with respect to the methodology. The rapid transferability of morphological parameters between scales: single plant, controlled net-house, and field conditions overlooked important technical details. Early statements about outcomes (abstract) of the use 2D and 3D parameters (abstract, line 30-32) to estimate early vigor were partially fulfilled further in the results section. Morphological parameters were not consistently used through different scales. 2-D, 3-D, and PSA parameters were utilized in single plant pots, PSA was only used in net-house, and PSA and EXG intensity used in field conditions. This added gaps of information, leaded to partial discussion and conclusions, and added confusion to reader. Not sure why 3-D parameters were not included in net-house and field conditions, UAV-based photogrammetry is widely used over large breeding nurseries and the price of software is not a strong argument.

  • We decided to use only PSA in the field and in the net-house competition experiments based on its high effectiveness in estimating early vigor, and the straightforwardness in its extraction.
  • We agree with the reviewer comment that it might be confusing for readers, we revised the Results section accordingly and added more information on experiments that were not included in the original manuscript:
  1. 3-D parameters (height and volume) were extracted in the field experiment as new Figure 9.
  2. New analysis evaluating the relation between weed biomass and the extracted morphological parameters was added in new Table 1.
  3. The vegetation index (ExG) data from the field experiment was eliminated and only morphological parameters are presented.
  4. In the net-house competition experiment PSA data was removed from the results and only biomass data is being presented.
  • Overall, we think that the new presentation of similar parameters for both experiments make the discussion more coherent and clearer.
  • The potential contribution of these parameters for weed-competitiveness is better demonstrated and our arguments about the advantages and disadvantages of each modelling approach are more founded.
  • Follow the reviewer comment, the current version of the manuscript focus on the morphological parameters without involving vegetation indices.

The outcomes of the most importance 3-D and 2-D morphological parameters defined at single plants pots does not directly transfer to the canopy level, where plant architecture and morphological parameters can be affected when neighbor plants are competing for light and resources. The assumptions about the direct transfer of 2-D and 3-D parameters between single plant to canopy level deserves a brief informative reference for reader.

  • As indicated above, we extracted the 3-D parameters from the field experiment under "canopy level" with large number of surrounding plants and inter-specific competition. This important aspects were also addressed in the revised Discussion section.

The use of excessive greenness index and thresholding are common methods used for detection of green and non-green objects in imagery and widely reported. Thresholding requires manual tuning, which significantly constrains scalability. In addition, the proposed method highly depends on efficient manual cleaning of weeds prior to UAV data collection. This point becomes mandatory for the pipeline to succeed by preventing misclassification of weeds as target crop, or it constraints the applicability only under idealistic low weed pressure conditions.

  • Indeed, weed-free plots are essential for the success of our phenotyping approaches. However, weed\crop classification is different research area that was not in the scope of this study. We added these two aspects were in the revised Discussion section and added relevant references.

Line 21. “The adequate phenotyping ability for early growth and competition was robust under controlled and field conditions.” But if I understood correct, the performance of the method is highly dependent on efficient manual cleaning before UAV data collection to ensure that PSA is not inflated by incorrectly including weeds as target crops.

  • See above.

Line 97. “iii) validate these phenotyping approaches under field conditions. Our results present a comprehensive comparison between 2-D and 3-D phenotyping approaches to detect early vigor related traits and facilitate high-throughput screening tools in cereal breeding for enhanced weed competitiveness.” This statement was not fulfilled, since there was not comparison between 2-D and 3-D in net-house and field conditions, 2-D and 3-D comparison was only carried out in single plant pots, not competing for light and resources.

  • As mentioned, 3-D parameters from the field experiments were added and present in the new Figure 9. We believe that the revised Discussion is improved and include better comparison between the methods.

Line 205. “Classification and segmentation of the vegetation pixels allowed the evaluation of triticale pixels exclusively for the PSA evaluations (Figure 4D).” In field conditions, it assumes a previous very efficient manual cleaning of the plots, then all greens pixels assumed as triticale.

  • See comment #1

Line 228. It is worthy to inform that results in 3.1 were based on single plant pots.

The text was revised as suggested.

Line 260. “Here, the X50 value”. X50 concept was not previously defined in the text. I assume this is a typo and it refers to x0 in equation 2.

  • Indeed, X0 was mistakenly mentioned as X50, and we revised the text accordingly.

Line 300. “Here, growth parameters were extracted from UAV and included PSA and ExG index value, which reflects the triticale growth by coverage area and reflectance intensity”. EXG spectral intensity is now included to characterize early vigor, but its use was not described in materials and methods, and it was not used in net-house scale, which shows an inconsistent utilization of parameters throughout different scales.

  • We agree with the reviewer comment that the use of ExG index may be confusing to the readers, we removed this aspect and focused on morphological parameters exclusively.

Line 337. X50 was not previously defined.

  •  

Line 348. “For these reasons, the 3-D modeling approach may not be suitable to apply in the breeding program, which prioritizes friendlier HTP tools [9,33].” UAV-based photogrammetry is widely used in large field nurseries in breeding programs. The price of the photogrammetry software is not a prior strong argument to discard 3-D parameters in net-house and field conditions in results and discussion sections. GUI Meshlab, OpenDroneMap are open source and free available alternatives to process ground or UAV-based RGB imagery.

  • We have revised the text accordingly.
  • As suggested, the price of 3-D software as a reason pro/con reasoning for using such phenotyping parameters was removed from the text. We added several other reasons that support the usage of the 2-D method for this specific phenotyping purpose.

Finally, we wish to thank you for handling the revision of our manuscript and the reviewers for their helpful comments. We truly believe that the current version of the manuscript is considerably improved and hope that you will find it suitable for publication in remote sensing.

Sincerely yours,

Dr. Ran Lati

Reviewer 2 Report

Comments and Suggestions for Authors

This study evaluated two image-driven plant phenotyping methods (i.e., 2D and 3D) to facilitate effective and accurate selection for early vigor in cereals (two triticale genotypes differentiating in vigor and growth rate). The general scopes refer to: evaluate the potential of these two image-based sensing methods, as a phenotyping tool for early vigor in cereals; identify and characterize key morphological traits that associate with weed-competitiveness and validate these phenotyping approaches under field conditions.

However, in the description of experimental research, it misses details of the general approach and its applicability. In addition, it is a good study, but I would propose a more detailed contextualization and a more critical result discussion.

General comments

Authors should explicit acronyms for each first time they are written (including all figures and tables captions).

The numbers of the affiliations of the authors must be reported in superscript.

In general, never use personal pronouns throughout the text but always impersonal ones.

Major comments

Keywords

Don't use the same words in the title as keywords.

Introduction

L49-54: here enter a definition/description of phenotyping to better contextualize the expressed concept.

From the aim of the article it is not clear how the analyses were performed (e.g., camera, UAV, etc.) and where (e.g., in field, greenhouse, etc.). It should be specified better.

Materials and Methods

L183: all the technical specifications of the UAV used are missing. Report, for example, a table similar to the following:

Details

Items

Specifications

drone

Weight

Dimensions

Max speed

Satellite positioning systems

Digital camera

Camera

Sensor Resolution

Image Sensor Type

Capture Formats

Still Image Formats

Video Recorder Resolutions

Frame Rate

Still Image Resolutions

GIMBAL

Control range Inclination

Stabilization

Obstacle detection distance

Operating environment

Remote Control

Operating Frequency

Max Operating Distance

Battery

Supported Battery Configurations

Rechargeable Battery

Technology

L204: Specify how the Outsu threshold works.

Results

Generally, all the statistical analyses realized should be specified in the materials and methods because otherwise the results are not clear to read (e.g., ANOVA).

Discussion

L318: The acronym HTP must be inserted much earlier in the text (i.e., introduction).

Conclusions and future perspective

For future studies, I would add the potential that multivariate modeling could have in terms of growth prediction, etc. In this study only basic statistical analyses were made.

Author Response

November 16th 2020

Dear Editor,

Remote Sensing

Re: Revision of “Image-based high-throughput phenotyping of cereals early vigor and weed-competitiveness traits (remotesensing-989904).

Thank you for your letter dated November 02, 2020, notifying us that our manuscript may be acceptable for publication in Remote sensing following major revision.

The manuscript was revised according to the comments made by the two reviewers (in Italics) as specified below.

In response to the comments of reviewer #2:

Authors should explicit acronyms for each first time they are written (including all figures and tables captions).

  • All acronyms were checked through the paper.

The numbers of the affiliations of the authors must be reported in superscript.

  •  

In general, never use personal pronouns throughout the text but always impersonal ones.

  • As suggested, we revised the text and removed personal pronouns.

Don't use the same words in the title as keywords.

  •  

L49-54: here enter a definition/description of phenotyping to better contextualize the expressed concept.

  • As suggested, we revised the text and added better description and contextualization of phenotyping ,with relevant references.

From the aim of the article it is not clear how the analyses were performed (e.g., camera, UAV, etc.) and where (e.g., in field, greenhouse, etc.). It should be specified better.

  • We have revised the objectives to make them clearer. Additionally, the headings of the different experiments, in the methods and the results sections, were edited and the exact platform (remote/proximal) and location (net-house/field) were spelled out (e.g., line 120).

L183: all the technical specifications of the UAV used are missing. Report, for example, a table similar to the following:

  • We revised the text and added the relevant data (see supplementary data Table S1).

L204: Specify how the Outsu threshold works.

  • As suggested, we added more details about the Outsu threshold methods. 

Generally, all the statistical analyses realized should be specified in the materials and methods because otherwise the results are not clear to read (e.g., ANOVA).

  • The exact statistical methods were spelled out in the text to match the method section. 

L318: The acronym HTP must be inserted much earlier in the text (i.e., introduction).

  • As suggested, HTP was mentioned in the introduction.

For future studies, I would add the potential that multivariate modeling could have in terms of growth prediction, etc. In this study only basic statistical analyses were made.

  • This is a good point, and the fact that multivariate analysis can improve future research was added to the discussion part. However, only basic statistic was used in this study as the main objective was to detect differences in morphological features and not detect/classify between cultivars.

Finally, we wish to thank you for handling the revision of our manuscript and the reviewers for their helpful comments. We truly believe that the current version of the manuscript is considerably improved and hope that you will find it suitable for publication in remote sensing.

Sincerely yours,

Dr. Ran Lati

Round 2

Reviewer 1 Report

Comments to Author:

Title: Image-based High-throughput Phenotyping of Cereals Early vigor and weed-competitiveness Traits.

Revision remotesensing-989904

Thank you for the detailed response to the review. Parameters are now more coherently compared through the different scales, but there is still some inconsistency. It still not clear to the reader why shoot convex hull and plant upper-width parameters (in 2.2 section) but there are not connected with the net-house and field evaluation. They may be applicable for individual plants but they are not previous report of the use of these parameters at commercial plant densities conditions (net-house or field), where plants growing close to each other rapidly overlap.

There are important methodological assumptions that are still overlooked in the discussion. “However, plant upper-width and height were even more sensitive parameters that showed the greatest differences between the genotypes through the net-house study, in terms of the magnitude of differences and repeatedly, respectively.” How is it possible to determine plant upper- width in the image once plants overlap to each other in net-house study? Again, there are no previous reports of that using simple color thresholding techniques.

Why PSA was removed from net-house results?

why does table 1 regression uses morphological parameters (predictors) of weed biomass? I'm not arguing that it is incorrect, but instead of that morphological parameters versus triticale biomass seems a more direct and clear evaluation of the outcomes.

For future research, I suggest to use  multivariate statistical techniques that  consider: i) the relative importance of each morphological parameter on early vigor, and ii)  potentially complement and add the partial contribution of morphological parameters to leverage early vigor determination (regression), rather than choosing one parameter over another. 

Author Response

November 19th 2020

Dear Editor,

Remote Sensing

Re: Revision of “Image-based high-throughput phenotyping of cereals early vigor and weed-competitiveness traits (remotesensing-989904).

Thank you for your letter dated November 19, 2020, notifying us that our manuscript may be acceptable for publication in Remote sensing following minor revision.

The manuscript was revised according to the comments made by the reviewers (in Italics) as specified below.

In response to the comments of reviewer #1:

  1. Thank you for the detailed response to the review. Parameters are now more coherently compared through the different scales, but there is still some inconsistency. It still not clear to the reader why shoot convex hull and plant upper-width parameters (in 2.2 section) but there are not connected with the net-house and field evaluation. They may be applicable for individual plants but they are not previous report of the use of these parameters at commercial plant densities conditions (net-house or field), where plants growing close to each other rapidly overlap.
  • We agree with the reviewer comment that some single-plant parameters such as upper-width are relevant for single pots and are not applicable for studies with plot level experiments. However, many breeding and genetic projects are being conducted under controlled conditions using single plants, and these parameters are relevant. Thus, we extract and analyzed these parameters and believe that they contribute and relevant for this paper. We added this important aspect to the discussion in the revised manuscript to further emphasize the rational of using these parameters estimation (lines 395-398).
  1. There are important methodological assumptions that are still overlooked in the discussion. “However, plant upper-width and height were even more sensitive parameters that showed the greatest differences between the genotypes through the net-house study, in terms of the magnitude of differences and repeatedly, respectively.” How is it possible to determine plant upper- width in the image once plants overlap to each other in net-house study?
  • This statement refers to the individual plant experiment. To avoid confusion, we revised the text (line 387-388).
  1. Again, there are no previous reports of that using simple color thresholding techniques.
  • It is not clear to us what the reviewer comment is about, and therefore we did not addressed this comment.
  1. Why PSA was removed from net-house results?
  • Follow the previous comments of the reviewer we removed the PSA to improve the consistency between the controlled and field experiments. We do believed that this figure is important and in the revised manuscript we added this data (New Figure 8) and revised the methods (line 171-173), results (Figure 8) and discussion sections (452-453) accordingly.   
  1. why does table 1 regression uses morphological parameters (predictors) of weed biomass? I'm not arguing that it is incorrect, but instead of that morphological parameters versus triticale biomass seems a more direct and clear evaluation of the outcomes.
  • One of the study objectives of the current study was to:" identify and characterize key morphological traits that associate with weed-competitiveness". By correlating the weed biomass (suppression\control) to the different triticale morphological parameters this objectives can be fulfill and we believe that this more relevant analysis.
  1. For future research, I suggest to use  multivariate statistical techniques that  consider: i) the relative importance of each morphological parameter on early vigor, and ii)  potentially complement and add the partial contribution of morphological parameters to leverage early vigor determination (regression), rather than choosing one parameter over another.
  • This is a good point. It was mentioned by reviewer #2 in the 1st round and we referred to this comment in the revised discussion (457-461). 

Finally, we wish to thank you for handling the revision of our manuscript and the reviewers for their helpful comments. We truly believe that the current version of the manuscript is considerably improved and hope that you will find it suitable for publication in remote sensing.

Sincerely yours,

Dr. Ran Lati

November 19th 2020

Dear Editor,

Remote Sensing

Re: Revision of “Image-based high-throughput phenotyping of cereals early vigor and weed-competitiveness traits (remotesensing-989904).

Thank you for your letter dated November 19, 2020, notifying us that our manuscript may be acceptable for publication in Remote sensing following minor revision.

The manuscript was revised according to the comments made by the reviewers (in Italics) as specified below.

In response to the comments of reviewer #1:

  1. Thank you for the detailed response to the review. Parameters are now more coherently compared through the different scales, but there is still some inconsistency. It still not clear to the reader why shoot convex hull and plant upper-width parameters (in 2.2 section) but there are not connected with the net-house and field evaluation. They may be applicable for individual plants but they are not previous report of the use of these parameters at commercial plant densities conditions (net-house or field), where plants growing close to each other rapidly overlap.
  • We agree with the reviewer comment that some single-plant parameters such as upper-width are relevant for single pots and are not applicable for studies with plot level experiments. However, many breeding and genetic projects are being conducted under controlled conditions using single plants, and these parameters are relevant. Thus, we extract and analyzed these parameters and believe that they contribute and relevant for this paper. We added this important aspect to the discussion in the revised manuscript to further emphasize the rational of using these parameters estimation (lines 395-398).
  1. There are important methodological assumptions that are still overlooked in the discussion. “However, plant upper-width and height were even more sensitive parameters that showed the greatest differences between the genotypes through the net-house study, in terms of the magnitude of differences and repeatedly, respectively.” How is it possible to determine plant upper- width in the image once plants overlap to each other in net-house study?
  • This statement refers to the individual plant experiment. To avoid confusion, we revised the text (line 387-388).
  1. Again, there are no previous reports of that using simple color thresholding techniques.
  • It is not clear to us what the reviewer comment is about, and therefore we did not addressed this comment.
  1. Why PSA was removed from net-house results?
  • Follow the previous comments of the reviewer we removed the PSA to improve the consistency between the controlled and field experiments. We do believed that this figure is important and in the revised manuscript we added this data (New Figure 8) and revised the methods (line 171-173), results (Figure 8) and discussion sections (452-453) accordingly.   
  1. why does table 1 regression uses morphological parameters (predictors) of weed biomass? I'm not arguing that it is incorrect, but instead of that morphological parameters versus triticale biomass seems a more direct and clear evaluation of the outcomes.
  • One of the study objectives of the current study was to:" identify and characterize key morphological traits that associate with weed-competitiveness". By correlating the weed biomass (suppression\control) to the different triticale morphological parameters this objectives can be fulfill and we believe that this more relevant analysis.
  1. For future research, I suggest to use  multivariate statistical techniques that  consider: i) the relative importance of each morphological parameter on early vigor, and ii)  potentially complement and add the partial contribution of morphological parameters to leverage early vigor determination (regression), rather than choosing one parameter over another.
  • This is a good point. It was mentioned by reviewer #2 in the 1st round and we referred to this comment in the revised discussion (457-461). 

Finally, we wish to thank you for handling the revision of our manuscript and the reviewers for their helpful comments. We truly believe that the current version of the manuscript is considerably improved and hope that you will find it suitable for publication in remote sensing.

Sincerely yours,

Dr. Ran Lati

Reviewer 2 Report

The manuscript in the present form can be published in Remote Sensing as the authors have been significantly improved it. Parts that better contextualize the work have been added (e.g., phenotyping). Both aim and discussion are now clearer.

I don’t need to review another version because I accept the work in the present form.

Best regards

Author Response

November 19th 2020

Dear Editor,

Remote Sensing

Re: Revision of “Image-based high-throughput phenotyping of cereals early vigor and weed-competitiveness traits (remotesensing-989904).

Thank you for your letter dated November 19, 2020, notifying us that our manuscript may be acceptable for publication in Remote sensing following minor revision.

The manuscript was revised according to the comments made by the reviewers (in Italics) as specified below.

In response to the comments of reviewer #1:

  1. Thank you for the detailed response to the review. Parameters are now more coherently compared through the different scales, but there is still some inconsistency. It still not clear to the reader why shoot convex hull and plant upper-width parameters (in 2.2 section) but there are not connected with the net-house and field evaluation. They may be applicable for individual plants but they are not previous report of the use of these parameters at commercial plant densities conditions (net-house or field), where plants growing close to each other rapidly overlap.
  • We agree with the reviewer comment that some single-plant parameters such as upper-width are relevant for single pots and are not applicable for studies with plot level experiments. However, many breeding and genetic projects are being conducted under controlled conditions using single plants, and these parameters are relevant. Thus, we extract and analyzed these parameters and believe that they contribute and relevant for this paper. We added this important aspect to the discussion in the revised manuscript to further emphasize the rational of using these parameters estimation (lines 395-398).
  1. There are important methodological assumptions that are still overlooked in the discussion. “However, plant upper-width and height were even more sensitive parameters that showed the greatest differences between the genotypes through the net-house study, in terms of the magnitude of differences and repeatedly, respectively.” How is it possible to determine plant upper- width in the image once plants overlap to each other in net-house study?
  • This statement refers to the individual plant experiment. To avoid confusion, we revised the text (line 387-388).
  1. Again, there are no previous reports of that using simple color thresholding techniques.
  • It is not clear to us what the reviewer comment is about, and therefore we did not addressed this comment.
  1. Why PSA was removed from net-house results?
  • Follow the previous comments of the reviewer we removed the PSA to improve the consistency between the controlled and field experiments. We do believed that this figure is important and in the revised manuscript we added this data (New Figure 8) and revised the methods (line 171-173), results (Figure 8) and discussion sections (452-453) accordingly.   
  1. why does table 1 regression uses morphological parameters (predictors) of weed biomass? I'm not arguing that it is incorrect, but instead of that morphological parameters versus triticale biomass seems a more direct and clear evaluation of the outcomes.
  • One of the study objectives of the current study was to:" identify and characterize key morphological traits that associate with weed-competitiveness". By correlating the weed biomass (suppression\control) to the different triticale morphological parameters this objectives can be fulfill and we believe that this more relevant analysis.
  1. For future research, I suggest to use  multivariate statistical techniques that  consider: i) the relative importance of each morphological parameter on early vigor, and ii)  potentially complement and add the partial contribution of morphological parameters to leverage early vigor determination (regression), rather than choosing one parameter over another.
  • This is a good point. It was mentioned by reviewer #2 in the 1st round and we referred to this comment in the revised discussion (457-461). 

Finally, we wish to thank you for handling the revision of our manuscript and the reviewers for their helpful comments. We truly believe that the current version of the manuscript is considerably improved and hope that you will find it suitable for publication in remote sensing.

Sincerely yours,

Dr. Ran Lati
